# Titanium Dental Implants: An Overview of Applied Nanobiotechnology to Improve Biocompatibility and Prevent Infections

**DOI:** 10.3390/ma15093150

**Published:** 2022-04-27

**Authors:** Rayane C. S. Silva, Almerinda Agrelli, Audrey N. Andrade, Carina L. Mendes-Marques, Isabel R. S. Arruda, Luzia R. L. Santos, Niedja F. Vasconcelos, Giovanna Machado

**Affiliations:** Centro de Tecnologias Estratégicas do Nordeste-Cetene, Av. Prof. Luiz Freire, 01, Cidade Universitária, Recife CEP 50740-545, PE, Brazil; rayane.silva@cetene.gov.br (R.C.S.S.); almerinda.agrelli@cetene.gov.br (A.A.); audrey.andrade@cetene.gov.br (A.N.A.); carina.marques@cetene.gov.br (C.L.M.-M.); isabel.arruda@cetene.gov.br (I.R.S.A.); luzia.santos@cetene.gov.br (L.R.L.S.); niedja.vasconcelos@cetene.gov.br (N.F.V.)

**Keywords:** osseointegration, biofilms, bone–implant interface, prostheses and implants, coating, surface modifications, nanotechnology

## Abstract

This review addresses the different aspects of the use of titanium and its alloys in the production of dental implants, the most common causes of implant failures and the development of improved surfaces capable of stimulating osseointegration and guaranteeing the long-term success of dental implants. Titanium is the main material for the development of dental implants; despite this, different surface modifications are studied aiming to improve the osseointegration process. Nanoscale modifications and the bioactivation of surfaces with biological molecules can promote faster healing when compared to smooth surfaces. Recent studies have also pointed out that gradual changes in the implant, based on the microenvironment of insertion, are factors that may improve the integration of the implant with soft and bone tissues, preventing infections and osseointegration failures. In this context, the understanding that nanobiotechnological surface modifications in titanium dental implants improve the osseointegration process arouses interest in the development of new strategies, which is a highly relevant factor in the production of improved dental materials.

## 1. Introduction

Oral health problems affect about 3.5 billion people worldwide, with an estimated 267 million people suffering from tooth loss [1]. Tooth loss is often associated with trauma, periodontal disease and caries, which may affect the individual’s health not only in aesthetic and social issues, but also by impairing chewing, speech, and increasing the risk of developing diseases [2,3]. One of the worst oral health conditions is the complete loss of dentition, known as edentulism, which although preventable, is still a common problem worldwide [4]. In this context, dental implants emerged as the main form of treatment for total or partial tooth loss, replacing mobile dentures that were anchored in remaining teeth or soft tissue, and which, as a consequence, caused their alteration over time [5].

The success of dental implants brought the possibility of restoring the dental functions and the health of the patient, being a market capable of moving around USD 4.6 billion globally [6]. Among the different materials found on the market, titanium implants are the most used due to their biocompatibility and low cost.

Titanium is a bioinert material, inducing little or no deleterious effect on the surrounding tissue. However, despite the description of several inherent advantages of the material, without adequate surface treatment, it ends up having a low integration with the bone and gingival tissue, which may lead to dental implant failures. These failures occur due to poor osseointegration, affecting the stability of the implant in the bone, which can lead to the establishment of infections and inflammatory processes in the peri-implant space [7]. To reduce such problems, different surface treatments are investigated to promote better osseointegration and prevent the formation of harmful bacterial biofilms. Nanotechnology has generated positive impacts in dentistry, being able to produce surfaces with a specific topography and chemical composition to improve the biocompatible characteristics of materials [8]. Commercial implants are already found with nanostructured surface modifications, such as SLActive^®^ (Straumann, Basel, Switzerland), which is reported to induce a faster response to osseointegration, and HAnane Surface^®^ (Promimic, Gothenburg, Sweden), which brings the titanium coating with nanohydroxyapatite and can stimulate the performance of osteoblasts and promote bone growth [9].

Surface bioactivation with biomolecules is also the subject of major investigations to ensure the long-term success of implants. When implanted, biomolecules from the blood or produced by the cells of the host tissue are initially adhered to the metal to later initiate the cellular anchoring itself [10]. Therefore, the bioactivation of materials with molecules that have biological properties can not only help cell adhesion, but also regulate their activity on the implant surface, inducing cell proliferation, migration and differentiation. In this context, this review discusses the different aspects involved in the successful osseointegration of titanium implants and the main surface treatments applied for the development of biomimetic surfaces used in these implants.

## 2. Titanium and Its Alloys

Titanium is the ninth most abundant metal and it was discovered by William Gregory in 1791. It presents itself in its pure form as a silver metal with unique physical-chemical characteristics, such as low density (4506 g/cm^3^) and high strength (590 MPa) [11]. Titanium can quickly react with oxygen and this provides resistance to corrosion on the metal’s surface because of the formation of an oxide layer on the metal’s surface. Studies with this metal are developed for the most diverse themes, such as applications in sports, pigments, jewelry, marine equipment, aerospace, and medical industries [12]. Concerning the dental industry, titanium and its alloys are known to be non-toxic and even more biocompatible than chromium-cobalt and stainless steel [13]. In addition, they are compatible with computed tomography (CT) and magnetic resonance imaging (MRI). These titanium biomaterials are the basis for the manufacture of prostheses and dental implants.

Due to the different properties observed in titanium forms, it has been verified that titanium oxide (TiO_2_) is the most reported in studies related to dental implants. TiO_2_ is formed by the high capacity of titanium metal to react with air forming hydroxyl and hydroxide groups, which gives it a high capacity for resisting corrosion. This oxide layer confers titanium, and its biocompatibility. TiO_2_ can be found in three different crystalline forms in ambient conditions: anatase, brookite, and rutile. The phase transitions are possible by performing heat treatment at the end of the synthesis. While brookite (that is arranged in orthorhombic geometry) is the most difficult to obtain, rutile and anatase (both presenting octahedral geometry) are easily formed [14]. The difference found between the rutile and anatase phases is due to distortions between the octahedral formed by TiO_6_. To obtain these structures, several methods can be used, from hydrothermal to electrochemical. Therefore, changes in the physicochemical parameters within the synthesis will lead to the preferential formation of one of the intended phases [15]. Thus, the phase directly affects the success of its use for applications in dentistry. Anatase is often associated with applications requiring osseointegration and, therefore, is the most used in dental implants.

Although other materials are found in the manufacture of dental implants according to their chemical composition, such as ceramics or polymers, at present, titanium is the material most commonly used [16]. Currently, six different types of titanium are available as implant biomaterials. Of these, four are grades of commercially pure titanium (CPTi) (Grade I, Grade II, Grade III, and Grade IV), which is 98–99.6% pure titanium, and two are titanium alloys (Ti-6Al-4V and Ti-6Al-4V—Extra Low Interstitial alloys). These grades differ in resistance to corrosion, strength, and ductility [17].

An ideal material for the fabrication of dental implants should be biocompatible and have adequate strength, toughness, and corrosion and fracture resistance. These properties are usually related to the oxygen residuals in the metal. Grade IV CpTi presents the highest oxygen content (0.4%) and consequently, excellent mechanical strength, which is why it is the most widely used type of titanium for dental implants [11].

Titanium alloys emerged with the interest of reducing device manufacturing costs and were considered a potential metallic material in the biomedical industry. The alloying elements added to titanium are largely divided into alphas (α) stabilizers, such as aluminum, oxygen, nitrogen, and carbon, and betas (β) stabilizers, such as vanadium, iron, nickel, and cobalt. Therefore, dental titanium alloys exist in three structural forms: α, β, or a combination of the two (α-β) [18]. The α-β combination alloy (Ti-6Al-4V) is the most used in dental applications [11]. It consists of 6% aluminum and 4% vanadium, and is highly strong and resistant to corrosion. Aluminum is an α-phase stabilizer. It increases the strength of the alloy and decreases its density. On the other hand, vanadium is a β-phase stabilizer [19]. Beta stabilizing elements are expensive when compared to α stabilizers [20]. Thus, replacing the common β stabilizers for cheaper substitutes is the current industry demand. On this matter, Fe is the most common element used to replace the β-stabilizing element because of its low cost and strongness. However, it has been reported that high temperatures promote the formation of intermetallic compounds, such as TiFe or Ti_2_Fe, which have a negative influence on the ductility and mechanical properties of alloys [21,22].

The surface of titanium implants is important because of their influence on interaction with the bone. The surface of the main materials used as dental implants (CpTi and Ti-6Al-4V) is composed of the oxide TiO_2_, which allows high resistance to corrosion with a clinical success rate of up to 99% [23,24]. Although aluminum remains the most important and commonly used α stabilizer, it was reported that it makes working and machining titanium alloys difficult [25]. The use of Ti-6Al-4V has been reported to have good biological acceptance [26,27]. However, small quantities of aluminum and vanadium are eventually released, which may induce an inflammatory process. Aluminum inhibits bone mineralization, leading to bone malformation and vanadium is cytotoxic and may induce allergic reactions [28,29]. This is why dental implants are more often made from CPTi. To prevent these biological problems, vanadium-free alloys, such as Ti-6Al-7Nb and Ti-5Al-2.5Fe, have been developed [17]. Furthermore, alloys composed of non-toxic elements, such as Nb, Ta, Zr, and Pd, are under development.

Recently, a new dentistry alloy based on the binary formulation of 83–87% titanium and 13–17% zirconium (Roxolid^®^, Straumann, Basel, Switzerland) has been developed. It has been related that it exhibits better tensile and fatigue strength characteristics compared to CpTi and Ti-6Al-4V. In vivo studies in animal models have shown bone integration of threaded zirconia implants comparable to that of titanium after insertion in different animal models [30,31,32].

As titanium is unaesthetic in the frontal area, ceramic implants have been constructed as dental implants [33]. Ceramics are known to present an inert behavior and good physical properties [16]. Firstly, it was used as a coating material for metal implants aiming to improve osseointegration. Over recent years, various forms of ceramic coatings have been used on dental implants. Bioactive ceramics, such as calcium phosphates and bioglasses, and inert ceramics, such as aluminum oxide and zirconium oxide are widely used in many medical, orthopedic, and dental applications [34].

Polymers have also been used as dental implant materials. Polymethylmethacrylate, polytetrafluoroethylene, polyethylene, polysulfone, and polyurethane are the most reported to be utilized in this matter [35]. Acting as a coating layer, polymeric materials are more easily manipulated and do not generate an electrolytic current as metals do. Although they are aesthetically pleasing, a lack of adhesion and immunologic reactions have been reported [16,36,37,38].

## 3. Osseointegration Process and Complications Associated with Dental Implants

Since the discovery of the phenomenon of osseointegration by P.I Branemark in 1952 based on some observations of experiments on rabbit fibia, titanium prostheses have been the object of study by several research groups in the world [39]. The osseointegration phenomenon can be broadly defined as a functional contact with sufficient stability between the prosthesis and the bone [40]. Especially for dental implants, the American Academy of Dental Implants, defined osseointegration in 1986 as “*Contact established without the interposition of non-bone tissue between normal remodeled bone and an implant entailing a sustained transfer and distribution of load from the implant to within the bone tissue*” [41].

In this way, a sequence of biological events is involved in osseointegration. After the insertion of the endosseous implant, immune and inflammatory responses occur, followed by angiogenesis and osteogenesis. In this process, physicochemical characteristics of the implant, such as topography and hydrophilicity, will allow the anchorage of blood proteins, forming interaction sites for cells through cell receptors called integrins. Then, cells, such as neutrophils, occupy the implant surface, and after 2 to 4 days, monocytes and macrophages arrive [42]. Such steps are essential for homeostasis thanks to the release of cytokines and growth factors which will induce collagen matrix deposition and initial bone tissue formation [43] (Figure 1). In addition, for osseointegration to be effective, the implant must have other specific characteristics such as adequate geometry, maximum contact between the implant and the tooth, roughness, usually in the range of 1.5 µm, the physical health of the host, and, more recently, changes in the surface of the implants [44]. These modifications can range from structural modifications on the implant surface to bioactivation with molecules capable of accelerating the osseointegration process and preventing complications related to the development of peri-implant mucositis and peri-implantitis.

### 3.1. Non-Infectious Complications

Even with technological advances in rehabilitation with oral implants, there are still failures that represent an increase in therapeutic time, causing additional costs and discomfort for the patient [45,46]. Several factors may lead to treatment failures, such as the occurrence of an inflammatory process in the peri-implant tissues and mechanical failures (fractures and loosening by non-infectious pathways) [47,48]. The main factors related to the variables that make oral rehabilitation treatment susceptible to failure can be divided by the patients and implant profiles [45,48,49,50].

#### 3.1.1. Patient’s Profile

It is important to clarify that not every patient can receive a prosthesis over implants for oral rehabilitation. Cases of pre-existing diseases, where patients had osteoporosis, diabetes, and hypertension, had a higher rate of treatment failure, and errors in the delay in healing that lead to a loose implant condition require a longer period of adaptation [45,51]. The continuous use of some drugs, such as those used by patients with autoimmune diseases, rheumatoid arthritis, and cancer treatment, can lead to failure in the dental implant, causing the body to recover at a slower pace [52].

Cases in the literature where carriers of genetic syndromes such as Down commonly present macroglossia and crossbite (reverse joint), unstable mechanical factors, and unfavorable occlusion, affect the osseointegration process and the success of implant therapy [52,53]. Smokers and those with a history of periodontal disease have significantly increased failure statistics in implant treatment [50,54].

Behavioral factors such as parafunctional habits associated with the anatomy itself, such as bone quality and quantity of the maxillary bones, are considered risk factors for failure as they generate occlusal overload and complications such as fractures in the implant. The correct positioning of the implant in these cases is essential to minimize stress and pressure on the site [47,55]. This is the case, for example, of people who suffer from bruxism, causing a strong pressure on the implant, which may yield and fracture [52].

Moreover, some systemic conditions influence implant osseointegration, such as cancer treatment, inflammatory bowel disease and osteoporosis. Chemotherapeutic agents used to treat cancer can induce vascular changes that culminate in bone poor nutrition as well as reduce the formation of the collagen matrix of bone tissue, both resulting in a weaker bone and leading to a possible reduction in the survival rates of dental implants [56,57]. High levels of pro-inflammatory cytokines are present in immune-mediated inflammatory bowel diseases, such as arthritis and ulcerative colitis, through the combination of the Toll-like receptor 4 (TLR4)/nuclear factor kappa B (NF-κB) signaling pathway, a crucial role in the regulation of inflammation. The TLR4/NF-κB receptor is a regulator of the processes of autophagy, oxidative stress and osteoclastogenesis [58,59,60]. In the case of osteoporosis, there is a loss of bone density due to both the aging process and the decrease in estrogen levels, leading to an increase in bone porosity and, consequently, increasing the risk of fractures [61].

#### 3.1.2. Implant Profile

The conical–cylindrical and conical–hexagonal shape designs are the most common among today’s implants and can vary in structural characteristics, such as 4–10 mm in length by 1 to 2 mm in diameter. Mini hexagonal implants with a length of 4–6 mm are the ones with the highest success rate and the shortest osseointegration time [45,62,63]. Some metallic oral implants can induce a hypersensitivity reaction, descriptions of paresthesias or dysesthesias in patients allergic to implant compounds, such as nickel and/or titanium, compound allergy symptoms including swelling, loss of taste, and a tingling sensation [62]. The influence of the anatomical location on the success of implants was evaluated by authors in implants placed in the maxilla and mandible. One evaluation criterion is the bone loss around the implant; in the maxilla the marginal bone loss was significantly higher than in the mandible in most of the patients [45,54,64] so the implant insertion site must be evaluated and taken into consideration in the treatment planning.

### 3.2. Infectious Complications

Microbial infections can make the osseointegration process difficult, leading, in extreme cases, to the loss of the implant [7]. The main infectious complications that lead to implant loss are known as peri-implantitis and peri-implant mucositis, which result from the patient’s immune response that leads to an inflammatory process in the mucosa and bone near the implant, both in association with the organized microorganisms in biofilm [65]. A frequency of around 30% of peri-implant diseases is estimated, with this rate being higher in smokers [66].

The oral microbiome is the second largest in the human body, with approximately 700 species of microorganisms such as bacteria, fungi, viruses, and protozoa that interact with each other synergistically, antagonistically, or even as signaling. These oral microorganisms adhere to each other and also to the biotic or abiotic matrix, grouping a finely organized community called biofilm [67].

Microorganisms in their different habitats can present in their free form, called planktonic microorganisms, or grouped in communities, the latter being their preferred form. The community of microorganisms attached to a surface is called biofilm. Microorganisms in a biofilm are protected by an exopolysaccharide (EPS) matrix formed by proteins, lipids, and extracellular DNA released from lysed cells. Up to 90% of the biofilm mass is made up of EPS [68].

Bacteria in biofilms can exchange genetic material via horizontal gene transfer, including mechanisms of conjugation, transformation, transduction, and membrane vesicles, acquiring new genes, including antibiotic resistance genes, which makes the treatment of infections more difficult [68]. In addition, due to the physical protection provided by EPS, microorganisms in a biofilm are more resistant to the action of antimicrobials and the host’s immune response, making them more difficult to eliminate and, therefore, facilitating the emergence of infectious processes [69].

Failure in dental implants is associated with periodontitis where there is a change in the microbial flora from a predominately Gram-positive non-motile, aerobic, and facultative anaerobic composition to a Gram-negative motile, anaerobic microbe. *Staphylococcus aureus* and coagulase-negative staphylococci are associated with peri-implant infections. As these microorganisms can adhere to titanium surfaces, they may be significant in the colonization of dental implants and subsequent infections [70]. It is concerning that biofilms are responsible for about 65% of diseases including peri-implantitis and periodontitis. Hence, the microbial attacks may cause dental implant failure [71].

Biofilms formed on the tooth surface are called dental plaque. Biofilm formation on teeth begins with bacterial adhesion to a film attached to the enamel. This film is constituted by salivary proteins which bacteria adhere to through surface molecules present on bacteria, especially lectins, that act as adhesins [71]. Once adhered, the biofilm formation process begins (as shown in Figure 2).

EPS allows microorganisms to remain attached to surfaces, protect them, and in addition, play a structural role that holds the microbiota together and gives the biofilm the characteristic mushroom shape [72]. EPS matrix components vary according to the microorganisms that are present in the biofilm and their formation is a key point for biofilm growth. In addition to this structural function, the EPS matrix protects the microorganisms from the biofilm. Biofilm formation is, therefore, a response of microorganisms to some inhospitable conditions [73], such as a lack of nutrients, changes in the environment’s pH, the presence of antimicrobial agents, and the action of the host’s immune system, among others [74].

The biofilm life stages are: 1. adhesion; 2. production of the EPS matrix; 3. microcolony formation; and 4. detachment and dispersal. Briefly, microorganisms in their planktonic form adhere to the biotic or abiotic surface through appendages such as flagellum, fimbriae, and pili, among others [75]. Initially, this adhesion is reversible, however, as other microorganisms attach, adhesion becomes irreversible. Microorganisms begin to produce the EPS matrix, and then maturation and three-dimensional growth of the biofilm occurs due to the multiplication of microorganisms within the matrix [69], reaching, usually, a mushroom shape [76]. The last stage is characterized by the detachment and dispersion of microorganisms from the biofilm, allowing these microbes to reach other sites far from the primary site of infection, where they will attach and initiate a new cycle of biofilm formation [77].

The mechanism by which microorganisms change from sessile to scattering cells involves a complex network of molecular changes based on the expression of genes that completely alter the phenotype of these microorganisms: genes expressing EPS and fimbriae are downregulated, while genes expressing the microbe’s phenotypic characteristics that are essential for its planktonic life, such as flagellum and chemotaxis, are upregulated [78].

The dispersion process is related to stress conditions within the microcolony, such as nutrient limitation, toxic waste accumulation, and O_2_ depletion, among others. Such conditions favor some microorganisms’ death, forming empty spaces in the microcolony center. Surviving microbes induce EPS dissolution and, through gene regulation processes, repress the expression of genes whose products favor their anchorage in the biofilm, such as fimbriae. At the same time, they begin to express factors that allow their locomotion and escape from the microcolony, as shown by flagellum, for example [78].

The organization and coordination of microorganisms in the biofilm are regulated by quorum sensing which is defined as an intra- and inter-species bacterial communication system based on the production and secretion of chemical signaling molecules called autoinducers which are responsible for the expression of certain genes. These molecules are only perceived by bacteria when there is a high microbial density and this mechanism plays an important role in microbial physiological processes, such as the expression of bioluminescence and virulence factors and resistance to antimicrobials [79].

In recent years, several studies have been developed to improve osseointegration and reduce microbial infections by modifying the surface of the dental implants, adding to them antibiotics and nanoparticles that bring antimicrobial and antibiofilm characteristics to the implants [80]. This subject will be better addressed in the next topics.

## 4. Nanotechnology for Promoting Osseointegration

### 4.1. Nanostructures on Titanium Surfaces

Surface modifications bearing nanostructures on titanium surfaces may provide properties to solve the main problems with dental implants’ fixation. Titanium and its alloys are bio-inert and have poor chemical bonding with bone at the early stage of the implantation [81]. For the optimal behavior of the implant, it expects a positive interaction with the extracellular matrix, which is on the nanoscale dimension and interacts with physiology cells also at the nanometric level. Artificial nanostructures on titanium surfaces are relevant for the cell–material interaction and enhance cell adhesion, proliferation, and differentiation, as well as the total bioactivity of the implant [82].

Titania nanotubes are the most explored nanostructure from titanium and its alloys due to the possibility of mimicking the bone structure and presenting a positive cellular response [83]. This approach is defined as a biomimetic surface, due to the reproduction of the original bone nanostructure with the formation of the artificial nanotubes on the implant surface [84]. Other structures, such as nanoparticles and nanopores, are still explored on implant surfaces. Table 1 summarizes the major nanostructures grown on titanium surfaces through different methods.

In recent studies, Park et al., (2021) [85] obtained nanoflowers of the TiO_2_ on the titanium surface by the hydrothermal method and explored the structure, composition, and morphology of the synthesized material. The material was deposited on titanium for 15 min on microwave radiation exposition, with a rutile phase formation and a super hydrophilic surface. The increase in surface area and high hydrophilicity improved the adhesion of the protein albumin on the implant surface. In another study, Wei and coworkers (2020) produced nanofibers of the poly (lactic-co-glycolic acid) (PLGA) loaded with the anti-inflammatory aspirin coatings on titanium by electrospinning. The aspirin was released from nanofibers for up to 60 days, avoiding peri-implant aseptic inflammation and the coatings promoting the osseointegration ability of the titanium implants, demonstrated with in vivo tests [86]. To achieve such morphological changes, some techniques were employed, which are addressed in the next topics.

### 4.2. Surface Modification Techniques (Methods)

The surface modifications on dental implants are made through some techniques, in which mechanical, chemical, electrochemical and layer addition methods stand out [6]. The surface modifications change the physicochemical properties, from morphology, wettability, and roughness, as well as confer new properties, such as antibacterial action and interaction with the cell environment [87].

#### 4.2.1. Mechanical Method

Mechanical treatments mainly modify the morphology of the implant surface and thus alter the roughness. Rougher surfaces provide greater adaptability of the implant to the environment. The most common mechanical methods are blasting and polishing, which use external forces to modify the metallic surface [88]. Granato et al., (2019) evaluated the effect of the chemical, blasting, and polishing treatment on the surface of the commercially pure Ti (grade II) and Ti-6Al-4V alloy (grade V), two titanium alloys used on dental implants, compared to the machined surface without treatment. The treatments presented similar results in the topography of both alloys, with improvement in the osseointegration during the first period of evaluation time [89].

#### 4.2.2. Chemical Methods

Among the chemical methods, which also change the morphology, there is the surface treatment with acid, which removes impurities and lamination marks from the metallic surface. Chemicals attack the substrate, increasing the roughness and contact area [90]. The conventional method uses a mixture of hydrochloric acid (HCl) and sulfuric acid (H_2_SO_4_) solutions, with a temperature around 50–70 °C in successive immersions and subsequent surface cleaning [91].

#### 4.2.3. Electrochemical Methods

Electrochemical modifications are based on oxidation and reduction reactions and work with electron transfers between electrodes. Usually, this procedure is performed in an electrochemical cell, with three electrodes composed of two inert materials (counter and reference electrodes) and the surface that will be modified (working electrode) [92]. In oxidation, a loss of matter that is deposited on an electrode that is being reduced occurs in one mass transfer. This kind of treatment includes anodization and electrodeposition [93,94]. Both are similar processes and consist of applying a potential or current difference to oxidize the material that will be deposited on the surface, not inducing corrosion or dissolution. To prevent the formation of biofilms, silver nanoparticles (AgNPs) can be immobilized on the implant’s surface by electrodeposition [95], which have shown active antibacterial activity and have a wide application in the medical field. AgNPs adhere to the bacterial wall, penetrate the bacteria and interact with cellular structures, such as peptides, biomolecules, and DNA, leading to bacterial dysfunction and death [86].

A specific electrochemical method with recent application in the biomedical field is plasma electrolytic oxidation (PEO), an anodic anodization at high voltage. Among the distinct features of the PEO method are the control of the thickness of the coating deposited on the titanium substrate, porosity, roughness, and the composition of the molecules inserted on the coating, i.e., ensuring the control of the properties desired with the surface modifications [96]. The PEO method has demonstrated that it is effective for the insertion of inorganic compounds in the porous layer of TiO_2_, elements such as Molybdenum [97] and e Niobium [98], that improve corrosion protection on the titanium surface. The method also improves the hydroxyapatite formation, which is usually aimed with PEO treatment, with mainly an electrolyte composition of a Ca and P precursor, in addition to the other molecules of interest. Santos-Coquillat et al., (2018) [99] utilized the PEO treatment to obtain a biocompatible surface to improve osseointegration, evaluating by in vitro and in vivo essays. The coating was composed for Ca/P in the ratio of the 2.0 and 4.0, and both presented good cell adhesion and proliferation of murine osteoblasts, with better bone-matrix mineralization for the ratio of ~1.8 of Ca/P.

#### 4.2.4. Layer-by-Layer Technique (LbL)

The modification of the implant surface by adding layers involves the intermolecular interaction between materials with opposite charges, making an electrostatic interaction occur or even making the layers interact by hydrogen bonding [100]. Monolayers can be added, but are not limited to these methods by Spray-drying, Dig-coating, or Spin-coating [93]. Chua et al., (2018) [101] functionalized the titanium surface with alternating layers of hyaluronic acid and chitosan, intercalated forming multilayers with polyelectrolytes (PEMs) and immobilizing the RGD peptide (arginine-glycine-aspartic acid) to increase the interaction with osteoblasts and mesenchymal cells, potentiating interactions according to the proposed changes.

## 5. Biomimetic and Bioactive Surfaces

Titanium-based implant devices are commonly used clinically and have been extensively examined through in vivo studies over the past 35 years to scientifically understand the workings of the implant–tissue interface. Thus, for the long-term success of these implants, osseointegration with the surrounding environment is one of the most desired factors when devices are implanted [102,103,104,105,106,107]. In this way, biomimetic surfaces have been developed aiming to achieve structural and biochemical characteristics that accelerate the integration process between the implant and the surrounding tissues, thus reducing the risk of inflammatory reactions and bacterial infections.

The coating with biocompatible molecules can stimulate cell adhesion, bone mineralization, and the formation of the extracellular matrix (ECM), accelerating the osseointegration process [104]. Molecules such as proteins, peptides, and mineral components (such as hydroxyapatite, growth factors, and antibiotics), are among the different molecules used to perform the functionalization of implants, with very satisfactory results having been achieved despite the challenges related to the immobilization and stability of these structures [108,109].

Collagen is the main component of ECM, and type I collagen is the most found in bone, constituting about 85% of the organic components, along with other ECM proteins such as laminas, fibronectin and vitronectin. These molecules form an adhesive layer through their sites of interaction with osteoblastic cell membrane integrins, which favors initial anchorage, as well as cell proliferation and differentiation [110,111]. In this context, the attempt to mimic an ECM-like microenvironment on the surface of implants has been a widely investigated strategy to recreate specific cellular anchorage sites. Chang (2016) [112] demonstrated in his study that coating the titanium surface with fibronectin increased the bone volume and stability of the dental implant, thus decreasing the treatment time. Likewise, studies have shown that type I collagen is efficient in promoting osseointegration by stimulating bone formation at cellular and molecular levels, positively regulating genes for osteocalcin (OC) and bone sialoprotein (BSP), which are related to differentiation osteoblasts and the matrix mineralization phase [113]. In addition, type I collagen induces fibroblast cell proliferation and positively regulates the gene expression of matrix metalloproteinases that are involved in ECM remodeling, which allows for better biological sealing of peri-implant tissues [114].

In general, the RGD motif, formed by a tripeptide sequence of arginine-glycine-aspartate (Arg-Gly-Asp), is responsible for the connection between ECM proteins and osteoblasts, and thanks to this function, the peptide sequence RGD has been investigated for its role in the functionalization of implant surfaces [111]. As they are found in cyclic and linear conformations, c-RGD and l-RGD, respectively, Heller (2017) [115] evaluated whether these structural differences between peptides would be able to generate different biological responses in the osseointegration process. According to their study, both forms are effective in stimulating the adhesion, proliferation, and differentiation of osteoblasts in experiments in vitro, but in the in vivo analysis, it was possible to observe a significant increase in vertical bone apposition with implants coated with the c-peptide RGD, showing that not only the specific sequence of peptides, but also their cyclic structure are important for the response to osseointegration. The literature includes a vast approach to the use of proteins and peptides for the functionalization of implants, with different combinations of molecules being evaluated. Vines (2012) [116] pointed out the importance of the biphasic constitution of ECM by organic and inorganic components, demonstrating that the formation of composites containing amphiphilic peptides and hydroxyapatite (HA) in a proportion of 66% was responsible for highlighting the osteogenic differentiation of mesenchymal cells. HA is the main mineral component that constitutes teeth and bones, presenting itself as an excellent candidate for use in the improvement of biomaterials. In addition to presenting good biocompatibility, bioactivity, and osteoconductivity, HA has a direct connection with natural bone, a characteristic defined as biointegration, which ends up inducing faster healing around the implant, although there are challenges related to the degradation of this structure with time [117]. Several studies investigate the deposition of HA on the surface of implants, evaluating not only the efficiency of surface coating methods, but also its incorporation with organic molecules, such as collagen, peptides, morphogenetic proteins, antimicrobial agents, and others [118,119,120,121,122].

Another group of biomolecules explored for their therapeutic use in tissue regeneration are growth factors (GF). GFs are proteins secreted by cells that have, among their properties, the ability to stimulate cell proliferation, migration, and differentiation, promoting bone repair after injury. Among the GFs with osteogenic action, we can mention the transforming growth factor-β (TGF-β), the bone morphogenetic proteins (BMPs), which belong to the TGF-β superfamily, the insulin-like growth factors (IGFs), and the factor of platelet-derived growth (PDGF) [123]. Among the more than 20 BMPs described, the literature reports that BMP-2, 6, and 9 have the best osteogenic potential, being shown that among these, BMP-9 has a greater capacity to promote cell differentiation and bone mineralization [124]. BMPs act in the differentiation of mesenchymal stem cells (MSCs) into osteoprogenitor cells, however, the emphasis given to BMP-9 is due to its osteogenic capacity not being negatively regulated by antagonists such as Noggin, which gives it a greater biological performance [125]. The study of the use of different types of BMPs in the functionalization of implants ranges from the use of isolated proteins to their combined use with organic and inorganic components, and although very promising results have been achieved, studies continue to advance to ensure a better controlled release and avoid the rapid leaching of these molecules from the implant’s surface, thus ensuring its long-term performance [126].

### Surface Graded Functionalized

In 1984, a group of Japanese researchers invented a new generation of composite materials, called functionally graded materials (FGMs) [127,128]. These materials consist of presenting specific characteristics that vary along its dimension, obtaining continuous gradual properties destined to act distinctly in the exposed regions. In this way, it is possible to obtain a hybrid material with several functions to be performed. This heterogeneous profile makes them more advantageous than homogeneous materials as they look like human structures, such as bones, teeth, and skin, ensuring better clinical performance. In this context, numerous research has been gaining prominence in various applications of FGMs, especially in dentistry [105,129].

Dental implants simultaneously form several interfaces with the biological system, considering the different tissues that the device travels through during and after the completion of the procedure. From the bottom up along the implant body, three interfaces are reported: (1) subgingival hard tissue, which is bone tissue, (2) transgingival soft tissue, and (3) supragingival soft tissue. Technically, the metallic implant attaches directly to the bone. The neck and implant platform adheres to the sulcular and junctional epithelium. Finally, there is the implant abutment that is in contact with the oral epithelium, which is visible in the oral cavity, where you will receive the crown. Each of these interfaces must be optimized to meet the different demands that the organism itself requires. Thus, hybrid implants or FGMs assume the profile desired by implant dentistry [130].

At the implant–hard tissue interface, osteogenic properties are extremely important to promote osseointegration, thus allowing the device to be efficiently and quickly fixed by the body, ensuring fracture resistance during occlusal loading [131]. In this case, the biomodification of the terminal end of the implant with collagen and hydroxyapatite may favor this process, as shown above. Subsequently, at the implant–transgingival soft tissue interface, the cell adhesion of keratinocytes and fibroblasts is quite desirable to ensure a cohesive and uniform epithelialization, to avoid bacterial infiltration. Therefore, the intermediate region of the device can be functionalized with collagen, peptides, or other biomolecules that favor the process of cell adhesion and growth, in addition to presenting compounds that inhibit bacterial growth. Finally, at the apical end of the implant, it is mandatory to present broad microbiological bioactivity and anti-adherence to prevent the growth of bacteria and fungi and, consequently, biofilm formation. This region of the implant is in contact with the oral mucosa and, therefore, is more likely to develop infectious processes. For the implant interfaces with the transgingival and supragingival soft tissue, colonization by bacteria is considered the main risk to trigger serious infections, such as peri-implantitis. This infectious process is usually accompanied by inflammation, inducing an immune reaction in the patient and, consequently, “bone loss” and implant “rejection”.

The challenge in obtaining hybrid implants or FGMs is what has motivated research to obtain concomitant biofunctions in different regions of the implant in the same metallic device, avoiding colonization by bacteria and promoting the osseointegration process. However, the studies carried out to date are restricted to homogeneously modifying the implant surface, just improving its interface with the bone. In the context of studies on the development of hybrid implants, knowledge is still rudimentary.

## 6. Conclusions

Here, we presented an overview of different surface treatments that are investigated for the development of high-performance titanium dental implants. We found that factors such as morphology and chemical composition are promising for the creation of biomimetic surfaces, resulting in implants that promote faster and more efficient osseointegration when compared to smooth surfaces. Nanostructured surfaces can generate a topography of porosity similar to bone and thus assist the bone healing process. The coating with biomolecules can stimulate cell adhesion, as well as differentiation, proliferation, and migration, favoring osseointegration. Finally, recent studies indicate that hybrid implants, with different types of modifications based on the microenvironment of insertion, are future challenges that may arise as new materials for the production of dental implants.

## Figures and Tables

**Figure 1 materials-15-03150-f001:**
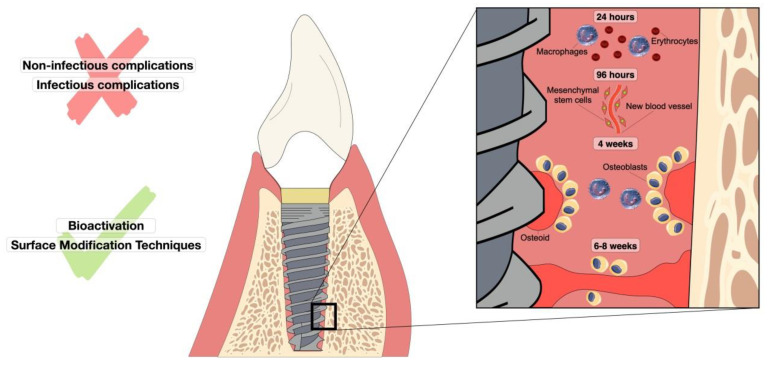
Representation of oral osseointegration events over time in a dental implant. The figure shows the sequence of cellular-level responses that occur after implant insertion for 24 h to approximately 8 weeks. Non-infectious and infectious complications are reported as factors that hinder osseointegration. Factors that improve this process are bioactivation and surface modification techniques.

**Figure 2 materials-15-03150-f002:**
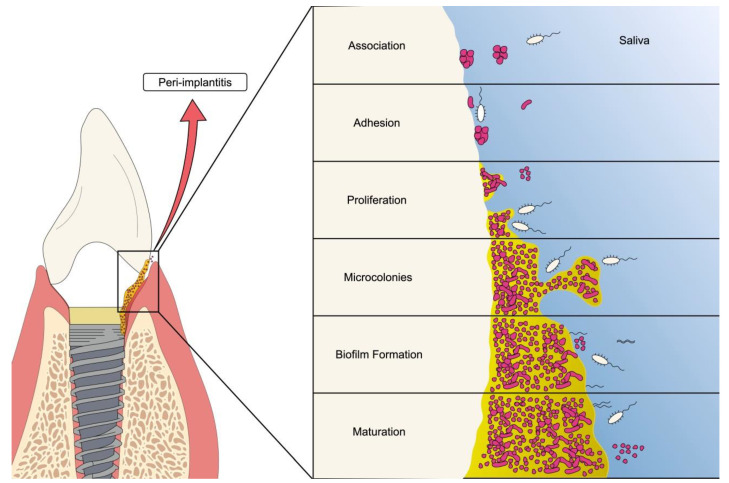
Schematic representation of oral biofilm formation on dental implants. The figure shows the different stages of bacterial biofilm formation ranging from adhesion to the establishment of the mature colony.

**Table 1 materials-15-03150-t001:** Nanostructured modifications on titanium and its alloys.

Nanostructure	Material	Method	Application	Ref.
Nanotubes	TiO_2_	Anodization	Experimental optimization	[79]
	TiO_2_/nano Brushite	Hydrothermal treatment/Anodization	Implant material/Bone regeneration	[80]
Silicate nanoparticle	TiO_2_	Acid etching/Electrospray deposition	Orthopedic and dental implants	[81]
Nanotubes/ Porous	Calcium phosphate-Sr-Si/TiO_2_	3D printing/ Anodization	Orthopedic and dental implants	[82]
Nanoparticles	Silver nanoparticles	Electrodeposition	Antibacterial property/Implant material	[83]
Nanowires	Zn-Ti	Acid etching/ Chemical treatment	Biocompatibility and antibacterial activity/ Implant material	[84]
Nanowire/coating	Na_2_Ti_3_O_7_/SrTiO_3_	Chemical treatment	Implant material	[85]
Nanofibers	Keratin/Ti	Mechanical treatment	Peri-implantitis/ Dental implants	[86]
Nanopores	TiO_2_	Chemical and electrochemical treatment	Biological integration/Dental implants	[87]
Nanotubes	TiO_2_/Hydroxyapatite/Chitosan	Electrochemical treatment	Dental implants	[88]

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
