# Peer review of "Titanium Dental Implants: An Overview of Applied Nanobiotechnology to Improve Biocompatibility and Prevent Infections"

_materials, 2022, doi:10.3390/ma15093150_

Round 1
Reviewer 1 Report
(I) The idea of this review paper is interesting. However, the surface modification of Ti and Ti alloys via plasma electrolytic oxidation should be discussed in this manuscript. Recently, several research groups reported that the incorporation of bioactive elements into the plasma electrolytic oxidation coatings is a very useful strategy to improve the bioactivity, biocompatibility, and corrosion properties of dental implants. The following examples should be cited in the manuscript.
- https://doi.org/10.1016/j.apsusc.2018.04.267
- https://doi.org/10.1016/j.corsci.2021.109764
- https://doi.org/10.1016/j.surfcoat.2019.125027
- https://doi.org/10.1016/j.surfcoat.2018.04.051
(II) The authors ought to go through the manuscript to revise complicated sentence structures as well as grammatical and spelling errors.
Author Response
Enclosed are the electronic files of the revised version of our manuscript, entitled “Titanium dental implants: an overview of applied nanobiotechnology to improve biocompatibility and prevent infections” (materials-1680897).
Our detailed response to the reviewer’s comments is as follows. We included a highlighted version of the revised manuscript to help guide throughout all the changes we made.
We hope that after revision, this version of the manuscript is suitable for publication.
Response to the Reviewer 1:
Comment (I): The idea of this review paper is interesting. However, the surface modification of Ti and Ti alloys via plasma electrolytic oxidation should be discussed in this manuscript. Recently, several research groups reported that the incorporation of bioactive elements into the plasma electrolytic oxidation coatings is a very useful strategy to improve the bioactivity, biocompatibility, and corrosion properties of dental implants. The following examples should be cited in the manuscript.
- https://doi.org/10.1016/j.apsusc.2018.04.267
- https://doi.org/10.1016/j.corsci.2021.109764
- https://doi.org/10.1016/j.surfcoat.2019.125027
- https://doi.org/10.1016/j.surfcoat.2018.04.051
Response: The text about plasma electrolytic oxidation method with cited references by the reviewer 1, was inserted in the manuscript on “Electrochemical methods” section, among the lines 413 to 427 .
Comment (II): The authors ought to go through the manuscript to revise complicated sentence structures as well as grammatical and spelling errors.
Response: As suggested by the reviewer, the English grammar has been completely revised. Please, check the revised manuscript.
Reviewer 2 Report
This is a very well written and comprehensive review of dental implants. The paper provided detailed manufacturing of surface types for increased osseointegration. There are some grammatical revisions that can further improve the paper. Overall excellent review!
Author Response
Response to the Reviewer 2:
Comment: This is a very well written and comprehensive review of dental implants. The paper provided detailed manufacturing of surface types for increased osseointegration. There are some grammatical revisions that can further improve the paper. Overall excellent review!
Response: We thank the reviewer for the kind comment. English has been revised throughout the manuscript, as suggested. Please check the revised manuscript.
Reviewer 3 Report
- How you choose the articles you used for your overview about titanium dental implants? Any search on specific databases?
- I am not sure the figures are yours. If not, their references must be cited
- In my opinion, reference 116 must be replaced
Author Response
Response to the Reviewer 3:
Comment: How you choose the articles you used for your overview about titanium dental implants? Any search on specific databases?
Response: Manuscripts were searched in electronic databases according to each topic covered in the review, based on titles, abstracts and relevance in the field of dental implants. The databases used were PubMed, ScienceDirect, Web of Science, CAPES Journal Portal and Google Scholar. The main terms used, alone or together, were Dental Implants, Titanium, Surface Modifications, Nanotechnology, Infections, Biofilms and Osseointegration. Relevant articles published in the last 10 years were selected and the general results were presented in a narrative form in the text or in tables.
Comment 2: I am not sure the figures are yours. If not, their references must be cited
Response: All figures in the review were prepared by the authors.
Comment 3: In my opinion, reference 116 must be replaced
Response: The reference has been replaced as suggested by the reviewer. The new reference is (line 502):
[Ref. 128] Sadollah, A.; Bahreininejad, A. Optimum Gradient Material for a Functionally Graded Dental Implant Using Metaheuristic Algorithms. J. Mech. Behav. Biomed. Mater. 2011, 4, 1384–1395.
Reviewer 4 Report
Review paper for English grammar.
Reference 81, page 701, Journal name is missing "Journal of Materials Science and Surface Engineering"
Comprehensive paper about titanium dental implants. Minor language and references revision required.
Author Response
Response to the Reviewer 4:
Comment: Review paper for English grammar.
Response: As suggested by the reviewer, the English grammar has been completely revised. Please check the revised manuscript.
Comment: Reference 81, page 701, Journal name is missing "Journal of Materials Science and Surface Engineering"
Response: As noted by the reviewer, the name of the journal was added to the reference (line 754 of the revised manuscript).
[Ref. 88] Dhatrak, P.; Shirsat, U.; Deshmukh, V. Fatigue Life Prediction of Commercial Dental Implants Based on Biomechanical Parameters : A Review. J. Mater. Sci. Surf. Eng. 2015, 3, 221–226.
Comment: Comprehensive paper about titanium dental implants. Minor language and references revision required.
Response: As suggested by the reviewer, the English grammar has been completely revised. Please check the revised manuscript.
Reviewer 5 Report
Dear Authors,
I read with great interest your review. Myself I am interested in the topic of implant osseointegration, mostly in systemic diseases. This review came in hand for me.
I would like to congratulate you for this comprehensive work. You worked really hard, you covered all the aspect that a paper like this should have.
From my perspective, there are few small things that my require improvements. The followings are:
Line 196: Here you should specify more systemic disease that influence implant osseointegration. Here we may mention about inflammatory bowel disease, osteoporosis, radio/chemotherapeutic regiment and so on. The followings may help you in improving this paragraph (doi: 10.1080/03602532.2019.1687511; doi: 10.1111/clr.13602)
Line 222: very nice that you add and described this important chapter.
Author Response
Response to the Reviewer 5:
Comment: Line 196: Here you should specify more systemic disease that influence implant osseointegration. Here we may mention about inflammatory bowel disease, osteoporosis, radio/chemotherapeutic regiment and so on. The followings may help you in improving this paragraph (doi: 10.1080/03602532.2019.1687511; doi: 10.1111/clr.13602)
Response: We are grateful for the reviewer's suggestion. We added in the revised manuscript (lines 225 - 239) a paragraph presenting the systemic diseases that negatively influence the osseointegration process of dental implants. Reference 54 (Chen, X.; Moriyama, Y.; Takemura, Y.; Rokuta, M.; Ayukawa, Y. Influence of Osteoporosis and Mechanical Loading on Bone around Osseointegrated Dental Implants: A Rodent Study. J. Mech. Behav. Biomed. Mater. 2021, 123, 104771, doi:https://doi.org/10.1016/j.jmbbm.2021.104771) has been removed and other references on the subject have been updated.
Round 2
Reviewer 1 Report
The authors have addressed all of my previous concerns, and their revisions have substantially improved the manuscript.